# Molecular Epidemiology of SARS-CoV-2 Omicron Sub-Lineages Isolated from Turkish Patients Infected with COVID-19

**DOI:** 10.3390/v15051066

**Published:** 2023-04-27

**Authors:** Murat Sayan, Ayse Arikan, Erdal Sanlidag

**Affiliations:** 1PCR Unit, Research, and Education Hospital, Faculty of Medicine, Kocaeli University, Kocaeli 41380, Turkey; murat.sayan@kocaeli.edu.tr; 2DESAM Research Institute, Near East University, Nicosia 99138, Cyprus; erdal.sanlidag@neu.edu.tr; 3Department of Medical Microbiology and Clinical Microbiology, Near East University, Nicosia 99138, Cyprus; 4Department of Medical Microbiology and Clinical Microbiology, Kyrenia University, Kyrenia 99320, Cyprus

**Keywords:** antiviral drug resistance, SARS-CoV-2, Omicron, protease inhibitors, RNA-dependent RNA polymerase, molecular epidemiology

## Abstract

Early detection and characterization of new variants and their impacts enable improved genomic surveillance. This study aims to evaluate the subvariant distribution of Omicron strains isolated from Turkish cases to determine the rate of antiviral resistance of RdRp and 3CLpro inhibitors. The Stanford University Coronavirus Antiviral & Resistance Database online tool was used for variant analyses of the strains uploaded to GISAID as Omicron (*n* = 20.959) between January 2021 and February,2023. Out of 288 different Omicron subvariants, B.1, BA.1, BA.2, BA.4, BE.1, BF.1, BM.1, BN.1, BQ.1, CK.1, CL.1, and XBB.1 were the main determined subvariants, and BA.1 (34.7%), BA.2 (30.8%), and BA.5 (23.6%) were reported most frequently. RdRp and 3CLPro-related resistance mutations were determined in *n* = 150, 0.72% sequences, while the rates of resistance against RdRp and 3CLpro inhibitors were reported at 0.1% and 0.6%, respectively. Mutations that were previously associated with a reduced susceptibility to remdesivir, nirmatrelvir/r, and ensitrelvir were most frequently detected in BA.2 (51.3%). The mutations detected at the highest rate were A449A/D/G/V (10.5%), T21I (10%), and L50L/F/I/V (6%). Our findings suggest that continuous monitoring of variants, due to the diversity of Omicron lineages, is necessary for global risk assessment. Although drug-resistant mutations do not pose a threat, the tracking of drug mutations will be necessary due to variant heterogenicity.

## 1. Introduction

The devastating impact of the COVID-19 pandemic, which has been going on for the last three years, continues. Since the beginning of the pandemic, various public health and social measures have been implemented to suppress transmission, improve patient management, and minimize the negative impact of the crisis on health systems, social services, and economic activities [1]. The pandemic has affected more than 750 million people; among those infected with COVID-19, 654,514,078 were recovered and 6,811,949 deceased, according to the World Health Organization (WHO) dashboard [2]. Moreover, more than 20 million active cases were reported, of which more than 40 thousand (2%) were critical [3].

For the last six months, Omicron (B.1.1.529) and its subvariants (Omicron BA.1, BA.2, BA.4, BA.5, and descendent lineages) have been the circulating variants of SARS-CoV-2, affecting many countries [4]. Although safe and effective vaccination programs have changed the course of the pandemic by reducing the number of cases and hospitalizations, there is still a need for effective prevention and treatment alternatives against COVID-19 because there are low-income countries with poor access to vaccines and drugs [5]. In addition, the obscurity of the duration of vaccine protection and the effectiveness of current vaccines against Omicron and other variants of SARS-CoV-2 necessitate new SARS-CoV-2 therapeutic approaches to limit morbidity and mortality in the different stages of the infection [6].

SARS-CoV-2 is a non-segmented positive-sense single-stranded RNA virus with a genome size of ~30 kb with a high replication capacity (10^−4^ to 10^−6^ substitutions per nucleotide per round of replication) [7]. The viral genome consists of more than six open reading frames (ORFs) that encode structurally (the spike (S), membrane (M), envelope (E), nucleocapsid (N)) and nonstructural proteins (NSP1-16) [8]. The S, E, M, N, replicase polyprotein, main protease (Mpro) and 3Chymotrypsin-like protease (3CLpro), papain-like protease (PLpro), RNA-dependent RNA polymerase (RdRp) (encoded by NSP12, NSP7, NSP8), NSPs, and transmembrane protease serine two are considered to be attractive drug targets for COVID-19 therapeutics to inhibit replication, viral maturation, and suppression of host innate immune responses [8,9,10,11,12]. Among those, the NSP-14 exoribonuclease (ExoN) and NSP 14-NSP10 complex have been shown to play a significant role in proofreading during replication and can evade host immune response [13,14]. Although nucleoside analogs are promising RdRp inhibitors, the efficiency of antivirals can be limited [13].

For the clinical management of COVID-19 cases, the U.S. Food and Drug Administration (FDA) granted authorization for SARS-CoV-2 targeting monoclonal antibodies, antiviral drugs, immune modulators, sedatives, and renal replacement therapies to treat mild-to-moderate and severe forms of COVID-19 [15]. Undoubtedly, antiviral treatment has a significant role in combating the COVID-19 pandemic. Remdesivir was the first FDA-approved therapeutic drug (October 22, 2020) which is an intravenous antiviral used in the treatment of adult and pediatric (>12 years) COVID-19 patients who are hospitalized and at high risk of hospitalization [16]. RDV inhibits RdRp by terminating the non-obligate RNA chain and inhibiting direct viral replication; thus, it reduces disease progression and promotes recovery [17]. Clinical trials showed that the efficacy of RDV in treating hospitalized patients with COVID-19 was 87% [18]. Another promising oral antiviral, molnupiravir, is a nucleoside analog given as emergency use authorization (EUAs) in December 2021 for treating mild-to-moderate infections that may progress to severe forms and unvaccinated COVID-19 cases [19]. Molnupiravir, which also targets RdRp, was authorized only for use in adults, and acts as a viral mutagen in the SARS-CoV-2 genome, accelerating the mutation rate and leading to the virus’s death [19,20]. Nirmatrelvir/ritonavir (NTS/r) (Paxlovid) is another drug that was developed in the fight against COVID-19 [19]. NTS/r is a highly recommended antiviral for treating non-severe and immunocompromised patients with an increased risk of progressing to severe COVID-19 and preventing hospitalization. NTS/r is an oral medication for adult and pediatric cases (>12 years old) that was developed for (EUA) by the FDA in December 2021. Paxlovid consists of nirmatrelvir, which is the 3CLpro inhibitor, and ritonavir, which is a cytochrome P450 inhibitor which boosts the exposure of nirmatrelvir, and its efficacy in reducing the risk of hospitalization and death was reported to be 89% in clinical phases [21]. Another promising candidate for treating mild-to-moderate COVID-19 patients is ensitrelvir. As a protease inhibitor, ensitrelvir has demonstrated favorable antiviral efficacy and an acceptable safety profile in clinical trials, even to Omicron variants [22].

While a decrease in the efficacy of various therapeutics was detected in previous studies [23,24], there is a need to investigate the effectiveness of antivirals in treating COVID-19. As the virus continues circulating and evolving, additional mutations may emerge that may cause drug resistance and other concerns. Therefore, the current study aims to evaluate the subvariant distribution of SARS-CoV-2 Omicron strains isolated from Turkish cases, and to determine the rate of antiviral drug resistance and the mutation patterns against SARS-CoV-2 RdRp and 3CLpro inhibitors in these strains.

## 2. Materials and Methods

### Genome Sequencing Retrieval, Variants Identification, and Antiviral Drug Resistance Analysis

The genome sequences of SARS-CoV-2 strains isolated from Turkey were obtained from the Global Initiative on Sharing All Influenza Data (GISAID) EpiCoV^TM^ database (https://www.epicov.org/epi3/frontend#41015d (accessed on 30 November 2022)) that provides initiative and primary data on coronaviruses associated with the COVID-19 pandemic. As Omicron (99.7%) was reported as the dominant SARS-CoV-2 variant of concern globally since December 2021, only the Omicron variant/subvariants were involved in the study [24]. The analysis was filtered for Variant of Concern GRA (B.1.1.529 + BA) Omicron complete genome sequences (~30 kb) from human hosts between January 2021 and February 2023, from the largest cities in Turkey with the highest populations including Istanbul, Ankara, Antakya, Karabuk, Izmir, Kayseri, Denizli, Adana, and Bursa. As the analysis only involved Omicron variants, the data were obtained between December 2021 and February 2023. 

Whole genome sequencing is becoming the most common approach for assessing the emerging variants that create antiviral resistance. A total of 20.959 entire genome sequences were downloaded as nucleotide sequences format (FASTA), and the alignments were used for variant identification and antiviral drug resistance analysis. Variant analysis and subvariant identification of the strains uploaded to GISAID as Omicron were performed with the Stanford Coronavirus Antiviral and Research Database (CoV-RDB) online tool (https://covdb.stanford.edu/ (accessed on 20 December 2022)) [25]. Mutations on SARS-CoV-2 gene regions spike, RdRp, nsp1, nsp2, PLpro, nsp4, 3CLpro, nsp6, nsp7, nsp10, nsp13-16, ORF3a, E, M, ORF6, ORF7a, ORRF7b, ORF8, N, and ORF10 were involved in the analysis [25].

Defining of SARS-CoV-2 variant/subvariants and the antiviral resistance mutations that target RdRp and 3CLPro inhibitors were assessed according to the mutation annotations given by CoV-RDB [25]. In this study, remdesivir (RDV) (Veklury, GS-5734, Gliead Sciences, Inc., Foster City, CA, USA) as RdRp inhibitor, NTV/r (Paxlovid/PF-07321332, Pfizer Labs, New York, NY, USA) and ensitrelvir (ENS) (Xocova/S-217622, Shionogi & Co., Ltd. Osaka, Japan) as 3CLpro inhibitors were assessed. Omicron sequences were explored with the consensus SARS-CoV-2 Omicron reference sequences, with mutations associated with RdRp and 3CLpro inhibitors. The resistance mutations included R285C, A449V, D484Y, V557L, S759A, V792I, E796G, C799F, C799R, E802A, and M924R, that are associated with the reduced activity to RdRp, were matched with GISAID mutations for RDV resistance. For the mutations related to the reduced activity against 3CLpro inhibitors, G15S, T21I, T45I, D48Y, M49I, M49T, L50F, G138S, F140L, N142D, N142L, N142S, G143S, S144A/E/L/P/R/T/V/W, C160F, M165T, E166A/G/K/V, L167F, H172Q/Y/T, H173V, V186G, R188S, Q189K, T190I, A191T/V, Q192A, and Q192E mutation patterns were involved and matched with the GISAID mutations for NTV/r and ENS resistance [25]. Antiviral drug resistance was interpreted for RdRp and 3CLPro inhibitors as medians ≥10-fold, 5–10-fold, 2.5–5, and <2.5-fold reductions in susceptibility, and gray cell as no susceptibility, according to the CoV-RDB protocols [25].

## 3. Results

In the current study, a new virtual phenotyping approach, CoV-RDB, identified Omicron variants and subvariants. According to our findings, Omicron variants were detected in Turkey as of December 2021. Within the given period, 288 different Omicron subvariants and descendent lineages were determined in a total of 20.959 Omicron sequences circulating in Turkey’s main cities. Remarkably, B.1, BA.1, BA.2, BA.4, BE.1, BF.1, BM.1, BN.1, BQ.1, CK.1, CL.1, and XBB.1 were the main determined subvariants of Omicron. Among these, BA.1 (*n* = 7293, 34.7%), BA.2 (*n* = 6463, 30.8%), and BA.5 (*n* = 4953, 23.6%) were the most common. While multiple descendent lineages were reported at higher rates mainly in BA.2, BA.5, BA.1, and BQ.1 subvariants, B.1, BA.4, BE.1, BF.1, BN.1, and XBB.1 subvariants were also reported to have descendent lineages. In this study, of the Omicron sequences exported to the GISAID, 0.04% (*n* = 9) were identified as unassigned sequences. The distribution of Omicron subvariants isolated in the Turkish population between December 2021 and February 2023 is shown in Table 1. The variant distribution including descendent lineages is provided in Appendix A. In addition, Figure 1 represents the circulation of dominant subvariants between December 2021 and February 2023. In Figure 2, the seasonal circulation of dominant Omicron subvariants is shown. According to these data, BA.1 followed by BA.2 subvariants were at a high rate in circulation, especially between January 2022 and March–April 2022. Moreover, a gradual decrease in the rate of Omicron subvariants over the months was observed.

Of the 20,959 Omicron sequences, antiviral resistance-associated mutations were determined in *n* = 150 (0.72%). The rate of the mutations against RdRp and 3CLpro inhibitors used against COVID-19 were reported to be 0.1% (*n* = 19/20.959) and 0.6% (*n* = 131/20.959), respectively. According to the subvariants and descendent lineages, antiviral resistance-associated mutations against RDV, NTS/r, and ENS were most frequently detected in BA.2 (*n* = 98, 65.3%) (including descendent lineages BA.2.10, BA.2.12, BA.2.17, BA.2.18, BA.2.23, BA.2.27, BA.2.3, BA.2.5, BA.2.58, BA.2.9, and BA.2.93), BA.1 (*n* = 26, 17.3%) (including descendent lineages BA.1.1, BA.1.1.1, BA.1.14, BA.1.14.1, BA.1.17, BA.1.17.2, BA.1.5, and BA.1.9), and BA.5.1 (*n* = 17, 11.3%) (including descendent lineages BA.5.2, BA.5.2.1, BA.5.2.2, and BA.5.6) variants. Reduced antiviral susceptibility was also determined in B.1.1.529 (0.7%), BA.4 (0.7%), BF.5 (1.3%), BQ.1 (0.7%), CK.1 (0.7%), and in unassigned variants at a rate of 2%. In total, 113 RdRp and 3CLpro inhibitor mutations were reported to reduce antiviral effectiveness in this study. Of these, 17 and 96 distinct mutation patterns were reported in RdRp and 3CLpro inhibitors-associated resistance, respectively. RdRp-related mutation patterns were reported as G15S, L50F/I/L/V, F140F/L, N142D/H/N/Y, G143C/G/R/S, C160C/F/S/Y, E166A/E/G/V, L167F/L, H172Q, R188S, T190I, A191A/E/G/P/S/T/V, A194A/P/S/T, D248E, T304I/N/S/T, and F305F/L. 3CLpro inhibitors-related mutation patterns were recorded as G15C/G/R/S, T21I/N/S/T, D48D/H/N/Y, M49M/I/T, L50C/F/L/I/S/V/Y, Y54C, F140F/I/L/V, G143C/G/R/S, S144L, C160C/F/S/Y, M165M/K/R/T, E166K/Q, L167L/F, H172R, A173T/V, V186A/D/G/V, R188S, Q189K, T190I, A191A/E/G/V, Q192K/L/P/R, A194A/P/S/T, P252L, T304I/N/S/T, and F305F/L in the current study. Table 2 shows the prevalence of SARS-CoV-2 PANGO lineages and their susceptibility rates to antiviral drugs (RdRp and 3CLpro inhibitors) with the mutation patterns. Additionally, no change in the rate of a specific mutation pattern over time was determined.

In the strains isolated from Turkish cases, RdRp resistance mutations were detected only in BA.1 (*n* = 2, 10.5%), BA.1.5 (*n* = 2, 10.5%), BA.2 (*n* = 10, 53%), BA.2.12 (*n* = 1, 5.2%), BA.2.58 (*n* = 1, 5.2%), BF.5 (*n* = 2, 10.5%), and in undefined variants (*n* = 1, 5.1%). On the other hand, reduced antiviral activity against 3CLpro inhibitors was detected in various Omicron subvariants except for BA.2.12, BA.2.18, and BF.5. Both resistance mutations were detected in five of the variants/sub-lineages with a rate of 65.3% (*n* = 98/150), including BA.1 (*n* = 10, 10%), BA.1.5 (*n* = 5, 5%), BA.2 (*n* = 77, 79%), BA.2.58 (*n* = 3, 3%), and unassigned variants (*n* = 3, 3%). A449A/D/G/V (*n* = 2, 10.5%) was the most common determined mutation pattern that reduced remdesivir activity, whereas T21I (*n* = 13, 10%), L50L/F/I/V (*n* = 8, 6%), and D48D/H/N/Y (*n* = 4, 3%) were the highest determined mutation patterns related with 3CLpro inhibitor resistance. The rates and the distribution of RdRp and 3CLpro-associated resistance mutations are shown in Table 2.

## 4. Discussion

Early detection and characterization of new variants and their impacts enable improved genomic surveillance. Additionally, developing antiviral drug resistance to viral infectious diseases due to the evolution of new forms is a growing concern. SARS-CoV-2 sequencing and the monitoring of novel variants and their effects is crucial in disease management; however, antiviral resistance to treatment regimens of SARS-CoV-2 has not been well-studied yet. In the current study, we used a virtual methodology to define the distribution of Omicron subvariants and descendent lineages circulating in Turkey to date and to identify the possible resistance-associated mutations of these strains to RdRp and protease inhibitors. Since December 2021, Omicron BA.1 (34.7%), BA.2 (30.8%), and BA.5 (23.6%) have been the dominant subvariants in Turkey, showing multiple varieties in their genome. Notably, although the XBB.1 subvariant was not the dominant variant, we observed a dramatic increase in XBB.1 within one month as the analysis has continued with 1000 new Omicron sequences for the last month (data not shown in Table 1, as Table 1 shows the rate of the variants after the study was completed). Besides subvariants, various descendent lineages, especially those belonging to BA.2, BA.5, BA1, BQ.1, B.1, BA.4, BE.1, BF.1, BN.1, and XBB.1 lineages, were also reported in Turkey according to our findings. As Omicron continues to circulate, the evolution and recombination of these variants led to the rise of the Omicron subvariants and descendent lineages worldwide [26]. In the current study, determining various Omicron subvariants and their descendent variants may explain why a new variant of concern has not emerged yet. Although there was a decline in the rates of BA.5 and its descendant lineages, these variants are still reported as dominant. Additionally, an increase in the rates of recombinant variants, mainly XBB.1, is remarkable [27]. The expanded diversity of Omicron variants reported in Turkey and the potential for the increase in the emergence of recombinant forms of Omicron revealed the necessity of continuous genomic surveillance to update vaccines and therapeutics used in the treatment.

Previous studies show NTV/r and ESV efficiency [28,29,30]. Therefore, the continued widespread use of these antiviral groups may trigger the development of drug resistance. Various clinical studies have shown resistance to NTV/r and ESV before [30,31,32]. Mutations such as T21I + E166V or L50F + E166V have been strongly associated with NTV/r resistance in vitro [32]. Additionally, P168S was described to diminish NTV and ESV by 5.1 and 6.8 folds, respectively [31]. In the current study, our findings also revealed the mutations that have been associated previously with reduced susceptibility to NTV/r, ENS, and RDV with amino acid changes D48D/H/N/Y, L50L/F/I/V, and A449A/D/G/V, respectively, for Omicron variants that have isolated in Turkish patients [30,31,32]. Throughout the pandemic, resistance mutations were detected in 0.73% of these strains circulating in Turkey. Reduced susceptibility was reported to RdRp (0.1%) and 3CLpro inhibitors (0.6%). The determined RDV, NTV/r, and ENS resistance against B.1, BA.1, BA.2, BA.4, BA.5, BF, BQ, and CK variants should raise concern in Turkey and support the need to monitor all presumable Omicron sequences for subvariants and antiviral resistance analysis. Mutation analysis may also help highlight mutation hot points. Notably, detecting multiple changes in RdRp and protease-inducing resistance to RDV, NTV/r, and ESV, especially in the BA.2 variant in Turkey, was remarkable. Although BA.2 is the second most frequently detected variant in circulation, the highest rate of antiviral drug resistance in this variant suggests that BA.2 should be monitored more closely. Conspicuously, the effectiveness of both antivirals in treating COVID-19 may be negatively affected due to the long-term use of these groups of antivirals. In addition, lower rates of RDV resistance to BA.1, BA.2, and BF strains suggest that despite its wide use, RDV remains a better option for treating cases infected with Omicron variants.

During the COVID-19 pandemic, bioinformatic tools have become more helpful in identifying emerging new variants and their potential impact on therapeutics, followed by whole genome sequencing [23]. CoV-RDB by Stanford University is a free, open-source virtual phenotyping system that provides lineage/variant, mutations, quality assessment, drug resistance comments, and susceptibility summaries in a short time and in an easily applicable way [25]. SARS-CoV-2 sequences can be accepted as FASTQ, FASTA formats, and spike amino acid mutations [25]. CoV-RDB is based on identifying nucleotide sequences using predetermined consensus SARS-CoV-2 sequences. A total of 12 RdRp and 61 3CLpro inhibitors resistance mutations are involved in the database at the global rate of > 1/10,000,000, and these mutations are updated monthly [25]. This new bioinformatic tool developed to identify antiviral drug resistance against COVID-19 involves RDV and NTV/r, ESV antivirals in the database that targets RdRp and 3CLpro, respectively [25]. Although molnupiravir has therapeutic potential for the treatment of COVID-19, it is most likely not included in the database as no reduction in antiviral susceptibility has yet to be reported [33]. The monthly update of the database allows the antivirals to be added, which develops resistance. However, not providing a version when updates are made may be a limitation of the CoV-RDB database.

## 5. Conclusions

In conclusion, the identification of SARS-CoV-2 Omicron variants, continuous monitoring of their subtypes, and the prediction of possible antiviral drug resistance mutations due to the general use of RDV, NTV/r, and ENS may support the development of new compounds or combinations in therapy. The CoV-RDB tool would be a good and valuable approach to identifying variants and their susceptibility patterns to therapeutics. However, due to its inability to identify recombinant variants, new web tools are needed to track recombinant subvariants of Omicron variants circulating in the population. During the pandemic, the variant and its effects can change very rapidly. Therefore, having versions of online tools can better reflect the period in which the study was conducted.

## Figures and Tables

**Figure 1 viruses-15-01066-f001:**
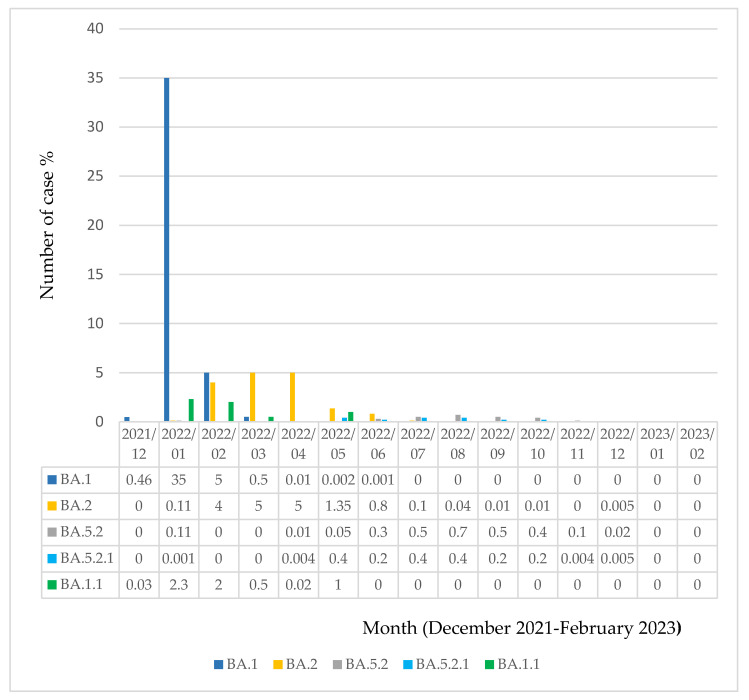
Circulation of dominant Omicron subvariants/descendent variants within 15 months. SARS-CoV-2 Omicron sub-lineages are BA.1, B.5.2, BA.2, BA. 5.2.1, and BA.1.1. Months involved were December 2021–February 2023. The figure presents the absolute number of cases as a percentage per month.

**Figure 2 viruses-15-01066-f002:**
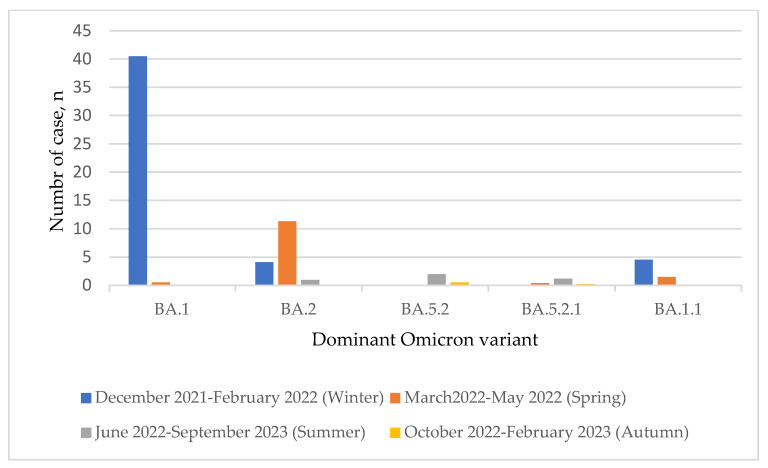
Seasonal circulation of dominant Omicron subvariants/descendent variants. SARS-CoV-2 Omicron sub-lineages are BA.1, BA. 2, BA.2, BA.2.1, and BA.1.1. Between December 2021 and February 2022, March 2022–May 2022, June 2022–September 2022, and October 2022–February 2023 were considered as Winter, Spring, Summer, and Autumn, respectively.

**Table 1 viruses-15-01066-t001:** The distribution of SARS-CoV-2 Omicron sub-lineages isolated in the Turkish population between December 2021 and February 2023.

SARS-CoV-2 Omicron Sub-Lineage	GISAID Sequence *n*, %
B.1	354 (1.68)
BA.1	7293 (34.7)
BA.2	6463 (30.7)
BA.4	93 (0.44)
BA.5	4953 (23.5)
BE.1	285 (1.35)
BF.1	21 (0.10)
BF.28	41 (0.19)
BF.5	138 (0.65)
BF.6	20 (0.09)
BF.7	49 (0.23)
BM.4.1	27 (0.12)
BN.1	62 (0.29)
BN.3.1	25 (0.11)
BQ.1	834 (3.97)
CK.1	27 (0.12)
CL.1	34 (0.16)
XBB.1	70 (0.33)
Others *	191 (0.91)
Unassigned **	9 (0.04)
TOTAL	20.959 (100)

* Others: Sub-lineages and descendent variants that were determined at very low rates. **: Amino acid substitutions that were not defined as subvariants. Note: Omicron subvariants/descendent variants are shown in alphabetical order in the table.

**Table 2 viruses-15-01066-t002:** Prevalence of SARS-CoV-2 Omicron sub-lineages and RdRp–3CLpro inhibitor resistance mutations in the study patients.

SARS-CoV-2 Omicron Sub-lineage	RdRp Inhibitor * a.a Substitutions	CoV-RDB, n (%)	3CLpro Inhibitor ** a.a Substitutions	CoV-RDB, *n* (%)
**B.1.1.529**	-		M49M/K/R/T	1 (0.8)
**BA.1**	A449V, C799C/G/R/S	2 (10.5)	G15S, L50L/F/I/V, A191A/E/G/T/V, F305F/L	8 (6.1)
**BA.1.1**	-		L50F, H172Q	2 (1.5)
**BA.1.1.1**	-		R188S	1 (0.8)
**BA.1.14**	-		G143G/C/R/S, A194A/P/S/T, T304T/I/N/S	2 (1.5)
**BA.1.14.1**	-		F140F/L, N142N/D/H/Y, E166E/A/G/V	2 (1.5)
**BA.1.17**	-		L167L/F, R188S	1 (0.8)
**BA.1.17.2**	-		A191A/E/G/V	1 (0.8)
**BA.1.5**	V557V/I/L, E802E/D	2 (10.5)	G15S, L50L/F/I/V, F140F/L, T190I	3 (2.3)
**BA.1.9**	-		L50L/F/I/V, C160C/F/S/Y, A191A/P/S/T, D248E	2 (1.5)
**BA.2**	R285R/C/G/S, A449A/D/G/V, D484Y, E796E/A/G/V, E802E/A/G/V	10 (52.6)	G15G/C/R/S, T21I/N/S, T45T/I/N/S, D48D/H/N/Y, M49I/M/T, L50C/L/F/I/S/Y/V, Y54C, F140F/I/L/V, N142S, G143G/C/R/S, S144L, C160C/F/S/Y, M165M/K/R/T, E166K/Q, L167L/F, A173T/V, H172R, V186V/A/D/G, R188S, Q189K, T190I, A191A/E/G/P/S/T/V, A194A/P/S/T, Q192Q/K/L/P/R, P252L, T304T/I/N/S, F305F/L	67 (51.2)
**BA.2.10**	-		T190I	1 (0.8)
**BA.2.12**	V792I	1 (5.3)	-	
**BA.2.17**	-		Q189K	1 (0.8)
**BA.2.18**	-		T45T/I/N/S, L50L/F/I/V	1 (0.8)
**BA.2.23**	-		Q189K	1 (0.8)
**BA.2.27**	-		F140F/L, A173A/D/G/V, A191A/E/G/V	1 (0.8)
**BA.2.3**	-		M49M/K/R/T, R188RS, Q189Q/E/K	2 (1.5)
**BA.2.5**	-		M49M/I, S144T, Q189K, T304I, F305F/L	3 (2.3)
**BA.2.58**	D484D/H/N/Y	1 (5.3)	D48D/H/N/Y, L50F, Y54C	3 (2.3)
**BA.2.9**	-		T45T/I/N/S, F140F/L, M165M/K/R/T, H172H/Q, F305F/L	5 (3.8)
**BA.2.93**	-		Y54Y/C/F/S	1 (0.8)
**BA.4.6**	-		L50F	1 (0.8)
**BA.5.1**	-		L50F	1 (0.8)
**BA.5.2**	-		T21I, M49I	9 (6.9)
**BA.5.2.1**	-		T21T/I/N/S, M49M/K/R/T, L50L/F/I/V, Y54C	5 (3.8)
**BA.5.2.2**	-		R188R/S, Q189Q/E/K	1 (0.8)
**BA.5.6**	-		T21I	1 (0.8)
**BF.5**	A449V	2 (10.53)	-	
**BQ.1**	-		T304I	1 (0.76)
**CK.1**	-		T21I	1 (0.76)
**Unassigned**	E802D	1 (5.26)	M49I, Q192L	2 (1.53)
**Total**		19 (100)		131 (100)

Abbreviations: WHO: World Health Organization; CoV-RDB: Stanford University Coronavirus Antiviral & Resistance Database; 3CLpro: 3 Chymotrypsin-like protease; RdRp: RNA-dependent RNA polymerase; a.a.: amino acid; -: not determined. * RdRp inhibitors; RVD: Remdesivir. ** 3CLpro inhibitors; NTV/r: Nirmatrelvir/ritonavir; ENS: Ensitrelvir. Note: Omicron subvariants/descendent variants and amino acid substitutions are shown in alphabetical order in the table.

## Data Availability

The datasets generated and analyzed during the current study are available from the corresponding author upon reasonable request.

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
