# Peer review of "Molecular Epidemiology of SARS-CoV-2 Omicron Sub-Lineages Isolated from Turkish Patients Infected with COVID-19"

_viruses, 2023, doi:10.3390/v15051066_

Round 1

Reviewer 1 Report

Dear authors, I recommend some minor changes in English writing.

Author Response

Manuscript ID: viruses-2323881

Title: Molecular Dynamics of SARS-CoV-2 Omicron Sub-lineages Isolated from

Turkish Patients Infected with COVID-19

Authors: Murat Sayan, Ayse Arikan *, Erdal Sanlidag

Dear Editor,

Please consider the corrections according to the reviewer’s comments. All the corrections was highlighted in red color within the main text.

Thank you for your time and assistance.

Yours Sincerely,

Ayse Arikan

Reviewer 1:

Dear authors, I recommend some minor changes in English writing.

We checked all the manuscript for the English writing.

Reviewer 2 Report

April 9, 2023

Title: Molecular Dynamics of SARS-CoV-2 Omicron Sub-lineages 2 Isolated from Turkish Patients Infected with COVID-19 

Authors: Murat Sayan, Ayse Arikan and Erdal Sanlidag

This manuscript reports a study on the subvariant distribution of Omicron strains isolated from Turkish cases using an online tool, Stanford Coronavirus Antiviral & Resistance Database. Authors showed the circulation of dominant Omicron subvariants/descendent, the distribution of Omicron sub-lineages, and the prevalence of the sub-lineages and resistance mutations, and determined the rate of antiviral resistance of RdRp and CLpro inhibitors. 

I agree with the authors statements that monitoring variant evolution together with drug-resistant mutations, is important for future global risk assessment, and find the value of this work. The manuscript is written clearly, and data is provided adequately. Thus, this work is publishable in Viruses. I suggest two minor points before publication.

(1)   Tables 1 and 2 are too long. Authors may reconsider the way to present these data, such as using graphs.

(2)   The title work “Molecular Dynamics” confused me and made me imagine physical chemistry research.

Author Response

(The authors gave the same response as above.)

Reviewer 3 Report

The manuscript entitled "Molecular Dynamics of SARS-CoV-2 Omicron Sub-lineages Isolated from Turkish Patients Infected with COVID-19" by Murat Sayan and colleagues analyzed SARS-CoV-2 sequences from the GISAID database to analyze the contribution of different Omicron subvariants in Turkey from January 2021 to February 2023. In addition, they used the Stanford University Coronavirus Antiviral and Research Database online tool to determine the frequency of mutations previously associated with resistance to remdesivir, nirmatrelvir/r and ensitrelvir among the omicron sublineages. While limited in scope, the manuscript is well written and the conclusions are supported by the findings. However, the manuscript leaves room for improvement:

Main concerns:

- the statement in lines 21-22 is misleading: The authors did not detect reduced susceptibility to specific antiviral drugs. This would require testing the actual viruses associated with the sequences in an antiviral assay. Please correct: " Mutations that have been associated previously with reduced susceptibility to remdesivir, nirmatrelvir/r and ensitrelvir were most frequently detected in BA.2 (51.3%). Please correct also in lines 257-260.

- it is unclear what the percentages in Table 2 are referring to. It appears they are the percentage of RdRp or 3CLpro mutations observed in a specific subvariant out of the total number of RdRp (19) and 3Clpro (131) resistance mutations. It would be much more informative for the reader to understand the percentage of sequences containing a specific resistance mutations within all sequences for a specific subvariant (prevalence). For example, how many % of the BA.2 sequences contain the T21I/N/S mutations? In addition, it would be very informative for the reader if they were changes over time in the prevalence of specific mutations.

- the very long tables are hard to read. Please consider presenting the data as a series of pie-charts.

Minor concerns: 

- line 50: is it replication capacity of replication fidelity?

-lines 78-80: this is misleading. Paxlovid consists of nirmatrelvir, which is the 3CLpro inhibitor, and ritonavir, which is a cytochrome P450 inhibitor and which boosts the exposure of nirmatrelvir.

-lines 63-88: suggest to group the RdRp inhibitors remdesivir and molnupiravir together, and the 3Clpro inhibitors nirmatrelvir and ensitrelvir

line 120: Foster City is situated in California (US), not Canada.

-Figure 1: Please clarify the y-axis: is this the absolute number of cases per month or a percentage. If percentage, is this per month or over the entire 15 month period. What is 100%?

-Figure 1 and 2 legend: please use the same nomenclature in the legend as in the figure itself.

- Figure 1 shows data from 12/2021 on, while the abstract states that sequences starting from January 2021 have been analyzed.

-Figure 2: any particular reason, why the 15 month period is not divided equally into 5 periods of 3 months?

-please consider including the rate of antiviral resistance mutations in the title. For example: "Rate of Resistance Mutations to Antiviral Drugs observed in SARS-CoV-2 Omicron Sub-lineages Isolated from Turkish Patients Infected with COVID-19"

Author Response

Manuscript ID: viruses-2323881

Title: Molecular Dynamics of SARS-CoV-2 Omicron Sub-lineages Isolated from

Turkish Patients Infected with COVID-19

Authors: Murat Sayan, Ayse Arikan *, Erdal Sanlidag

Dear Editor,

Please consider the corrections according to the reviewer’s comments. All the corrections were highlighted in red color within the main text.

Thank you for your time and assistance.

Yours Sincerely,

Ayse Arikan

Reviewer 3:

The manuscript entitled "Molecular Dynamics of SARS-CoV-2 Omicron Sub-lineages Isolated from Turkish Patients Infected with COVID-19" by Murat Sayan and colleagues analyzed SARS-CoV-2 sequences from the GISAID database to analyze the contribution of different Omicron subvariants in Turkey from January 2021 to February 2023. In addition, they used the Stanford University Coronavirus Antiviral and Research Database online tool to determine the frequency of mutations previously associated with resistance to remdesivir, nirmatrelvir/r and ensitrelvir among the omicron sublineages. While limited in scope, the manuscript is well written and the conclusions are supported by the findings. However, the manuscript leaves room for improvement:

Main concerns:

- the statement in lines 21-22 is misleading: The authors did not detect reduced susceptibility to specific antiviral drugs. This would require testing the actual viruses associated with the sequences in an antiviral assay. Please correct: " Mutations that have been associated previously with reduced susceptibility to remdesivir, nirmatrelvir/r and ensitrelvir were most frequently detected in BA.2 (51.3%). Please correct also in lines 257-260.                                                                                                                                     

 Necessary corrections have been done.

- it is unclear what the percentages in Table 2 are referring to. It appears they are the percentage of RdRp or 3CLpro mutations observed in a specific subvariant out of the total number of RdRp (19) and 3Clpro (131) resistance mutations. It would be much more informative for the reader to understand the percentage of sequences containing a specific resistance mutations within all sequences for a specific subvariant (prevalence). For example, how many % of the BA.2 sequences contain the T21I/N/S mutations? In addition, it would be very informative for the reader if they were changes over time in the prevalence of specific mutations.                                                                                                                                                                                   

Table 2 has been revised. no change in the rate of a specific mutation pattern over time was determined.

This information has been added to the result section.

- the very long tables are hard to read. Please consider presenting the data as a series of pie-charts.

We presented only the subvariants in the tables. We also provided all the subvariants and their descendent variants in the supplementary files.

Minor concerns: 

- line 50: is it replication capacity of replication fidelity?                                                                                          It is replication capacity

-lines 78-80: this is misleading. Paxlovid consists of nirmatrelvir, which is the 3CLpro inhibitor, and ritonavir, which is a cytochrome P450 inhibitor and which boosts the exposure of nirmatrelvir.                          The sentence has been corrected according to your recommendation.

-lines 63-88: suggest to group the RdRp inhibitors remdesivir and molnupiravir together, and the 3Clpro inhibitors nirmatrelvir and ensitrelvir                                                                                                                          RdRp inhibitors and 3CLpro inhibitors have been given in the same groups, respectively.

line 120: Foster City is situated in California (US), not Canada.                                                               

 A necessary correction has been done.’ Foster City, California, US

-Figure 1: Please clarify the y-axis: is this the absolute number of cases per month or a percentage. If percentage, is this per month or over the entire 15 month period. What is 100%?                                           Figure 1 presents the number of cases as a percentage per month.

-Figure 1 and 2 legend: please use the same nomenclature in the legend as in the figure itself.                        The same nomenclature has been used in the figure and its legend.

- Figure 1 shows data from 12/2021 on, while the abstract states that sequences starting from January 2021 have been analyzed.                                                                                                                                                         We performed the analysis between January 2021 and February 2023 however, Omicron variants were detected after December 2021 in Turkey. This explanation has been written in the main text in the result section. ‘According to our findings, Omicron variants have been detected in Turkey as of December 2021’

-Figure 2: any particular reason, why the 15 month period is not divided equally into 5 periods of 3 months?                                                                                                                                                                                            We tried to divide 15 months according to the seasons.

-please consider including the rate of antiviral resistance mutations in the title. For example: "Rate of Resistance Mutations to Antiviral Drugs observed in SARS-CoV-2 Omicron Sub-lineages Isolated from Turkish Patients Infected with COVID-19"                                                                                                       Thank you for your recommendation. At first, we also thought to include resistance mutations in the title. However, since we have included the frequency of omicron subvariants in the study, we preferred to use a more general title that includes prevalent and drug resistance analysis.